# Observation of unusual topological surface states in half-Heusler compounds LnPtBi (Ln = Lu, Y)

Z.K. Liu[1], L.X. Yang[2], S.-C. Wu[3], C. Shekhar[3], J. Jiang[1,4], H.F. Yang[5], Y. Zhang[4], S.-K. Mo[4], Z. Hussain[4], B. Yan[1,3], C. Felser[3] & Y.L. Chen[1,2,6]

Topological quantum materials represent a new class of matter with both exotic physical phenomena and novel application potentials. Many Heusler compounds, which exhibit rich emergent properties such as unusual magnetism, superconductivity and heavy fermion behaviour, have been predicted to host non-trivial topological electronic structures. The coexistence of topological order and other unusual properties makes Heusler materials ideal platform to search for new topological quantum phases (such as quantum anomalous Hall insulator and topological superconductor). By carrying out angle-resolved photoemission spectroscopy and *ab initio* calculations on rare-earth half-Heusler compounds LnPtBi (Ln = Lu, Y), we directly observe the unusual topological surface states on these materials, establishing them as first members with non-trivial topological electronic structure in this class of materials. Moreover, as LnPtBi compounds are non-centrosymmetric superconductors, our discovery further highlights them as promising candidates of topological superconductors.

[1] School of Physical Science and Technology, ShanghaiTech University and CAS-Shanghai Science Research Center, Shanghai 201203, China. [2] State Key Laboratory of Low Dimensional Quantum Physics, Department of Physics and Collaborative Innovation Center for Quantum Matter, Tsinghua University, Beijing 100084, China. [3] Max Planck Institute for Chemical Physics of Solids, D-01187 Dresden, Germany. [4] Advanced Light Source, Lawrence Berkeley National Laboratory, Berkeley, California 94720, USA. [5] State Key Laboratory of Functional Materials for Informatics, SIMIT, Chinese Academy of Sciences, Shanghai 200050, China. [6] Physics Department, Oxford University, Oxford OX1 3PU, UK. Correspondence and requests for materials should be addressed to Y.L.C. (email: Yulin.Chen@physics.ox.ac.uk).

Topological quantum materials, a new class of matter with non-trivial topological electronic structures, has become one of the most intensively studied fields in physics and material science due to their rich scientific significance and broad application potentials[1,2]. With the worldwide effort, there have been numerous materials predicted and experimentally confirmed as topologically non-trivial matter (including topological insulators[3,4], topological crystalline insulators[5,6] and three-dimensional (3D) topological Dirac and Weyl semimetals[7–11]). However, there is a big family of materials—the Heusler compounds—although being theoretically predicted to be topologically non-trivial back in 2010 (refs 12–14), the non-trivial topological nature has never been experimentally confirmed up to date.

The ternary semiconducting Heusler compounds, with their great diversity ($\sim 500$ members, $>200$ are semiconductors) give us the opportunity to search for optimized parameters (for example, spin–orbit coupling (SOC) strength, gap size and so on) across different compounds—which is critical not only for realizing the topological order and investigating the topological phase transitions[12], but also for designing realistic applications. In addition, among the wealth of Heusler compounds, many (especially those containing rare-earth elements with strongly correlated $f$ electrons) exhibit rich interesting ground state properties, such as magnetism[15,16], superconductivity[17–19] or heavy fermion behaviour[20]. The interplay between these properties and the topological order makes Heusler compounds ideal platforms for the realization of novel topological effects (for example, exotic particles including image monopole effect and axions and so on), new topological phases (for example, topological superconductors[21,22]) and broad applications (see ref. 23 for a review).

Rare-earth half-Heusler compounds LnPtBi (Ln = Y, La and Lu) represent a model system recently proposed that can possess topological orders with non-trivial topological surface states (TSSs) and large band inversion[12,24]. Moreover, due to the lack of the inversion symmetry in their crystal structure, non-centrosymmetric superconducting LnPtBi compounds (Tc = 0.7[17], 0.9[18] and 1.0 K[19] for Ln = Y, La and Lu, respectively) may also host unconventional cooper pairs with mixed parity, making them promising candidates for the investigation of topological superconductivity and the search for Majorana fermions[25].

However, despite the great interests and intensive research efforts in both theoretical[12,13,26] and experimental[27–29] investigations, the topological nature on LnPtBi remains elusive. A previous angle-resolved photoemission spectroscopy (ARPES) study has reported metallic surface states[27] with apparently different dispersion shapes and Fermi surface (FS) crossing numbers from the predicted TSSs in LnPtBi compounds[12,13], making the topological nature of LnPtBi controversial.

In this work, by carefully performing comprehensive ARPES measurements and *ab initio* calculations, we resolve this unsettled question. We observe the non-trivial TSSs with linear dispersions in half-Heusler compounds LuPtBi and YPtBi (on the Bi-and Y-terminated (111) surface, respectively); and remarkably, in contrast to many topological insulators that have TSSs inside their bulk gap[1,3,30], the TSSs in LnPtBi show their unusual robustness by lying well below the Fermi energy ($E_F$) and strongly overlapping with the bulk valence bands (similar to those in HgTe[31–33]). In addition to the TSSs, we also observe numerous metallic surface states crossing the $E_F$ with large Rashba splitting, which not only makes them promising compounds for spintronic application, but also provides the possibility to mediate topologically non-trivial superconductivity in the superconducting phase of these compounds.

## Results

**Basic physical properties of LnPtBi.** A crystallographic unit cell of LnPtBi is shown in Fig. 1a, which comprises of a zinc-blend unit cell from Bi and Pt atoms and rocksalt unit cell from Bi and Ln. For ARPES measurements, the LnPtBi single crystals were cleaved *in situ* in the ultra-high vacuum measurement chamber, resulting in either (111) or (001) surfaces. The unit cell along the (111) cleavage surface is illustrated in Fig. 1b and the corresponding hexagonal surface Brillouin zone (BZ) is shown in Fig. 1d, which could be viewed as the projection of the bulk BZ (Fig. 1c) of LnPtBi along the [111] direction (Fig. 1d).

The unit cell along the [111] direction consists of alternating Ln, Pt and Bi layers (Fig. 1b). As there are fewer chemical bonds to break between Ln and Bi layer (two comparing to three between Ln and Pt or Pt and Bi layers) and the Ln–Bi layer distance is twice as large as the Ln–Pt or Pt–Bi layer spacing (see Fig. 1b), it is more energetically favourable to cleave the material between Ln and Bi layers. In the discussion of the main text, we will focus on the electronic structure of Bi-terminated (111) surface LuPtBi and Y-terminated (111) surface of YPtBi. ARPES results (as well as *ab initio* calculations) along (001) cleavage surfaces of LuPtBi and YPtBi are presented in Supplementary Information.

The core level photoemission spectra of LuPtBi is shown in Fig. 1e, from which the characteristic Bi $5d$ and Lu $4f$ doublets are clearly observed. The large spectral weight of Bi peaks over the Lu peaks in the (111) surface clearly indicates its Bi termination. The broad area FS mapping covering multiple BZs in Fig. 1f also illustrates the hexagonal symmetry (with the correct lattice parameters) resulting from the (111) cleaved surface.

**General electronic structure of LuPtBi (111) surface.** In Fig. 2, detailed electronic structures of LuPtBi within a surface BZ are illustrated. From the FS maps (Fig. 2a–c) and 3D spectral intensity plots (Fig. 2d,e) around both the $\bar{\Gamma}$ point and BZ boundary ($\bar{K}$ and $\bar{M}$ points), there are clearly multiple bands crossing $E_F$, forming a twin hexagonal hole pockets at $\bar{\Gamma}$ and complex electron pockets at $\bar{K}$ and $\bar{M}$, both of which show clear Rashba splitting. Around $\bar{\Gamma}$, there is another pair of double $\Lambda$ shape hole bands just below $E_F$. These features broadly agree with the previous ARPES report[27] (also see Supplementary Figs 1 and 2 and Supplementary Note 1 for the detailed discussions on the features near $\bar{\Gamma}$). However, in this work, with the high instrument resolution and data statistics we successfully observed a critical additional X shape band dispersing between 0.4 and 0.8 eV with the band crossing point (that is, Dirac point) at $E_b \sim 0.5$ eV at the $\bar{\Gamma}$ point—which is the long-sought-after TSSs, as we will discuss in details below.

**Surface states on LuPtBi (111) surface.** To help understand the origin of these electronic states, we carried out band structure calculations of Bi-terminated LuPtBi (111) surface with two different methods (Fig. 3a,b, see the 'Methods' section for more details), and both agree well with the measurement and clearly reproduce the X shape TSS observed in our measurements (Fig. 3c–f). In Fig. 3a, we first employed a slab model for the *ab initio* calculations (method one). This method, which takes into account the charge density redistribution due to surface potential modification by *ab initio* calculations, can describe all surface states including TSSs and those from the trivial dangling bonds. To further identify the TSS, we carried out another method (Fig. 3b) by calculating the k-resolved local density of states of a semi-infinite surface using the recursive Green's function (method two)[34] constructed from Wannier function-based tight-binding parameters extracted from the bulk material[35].

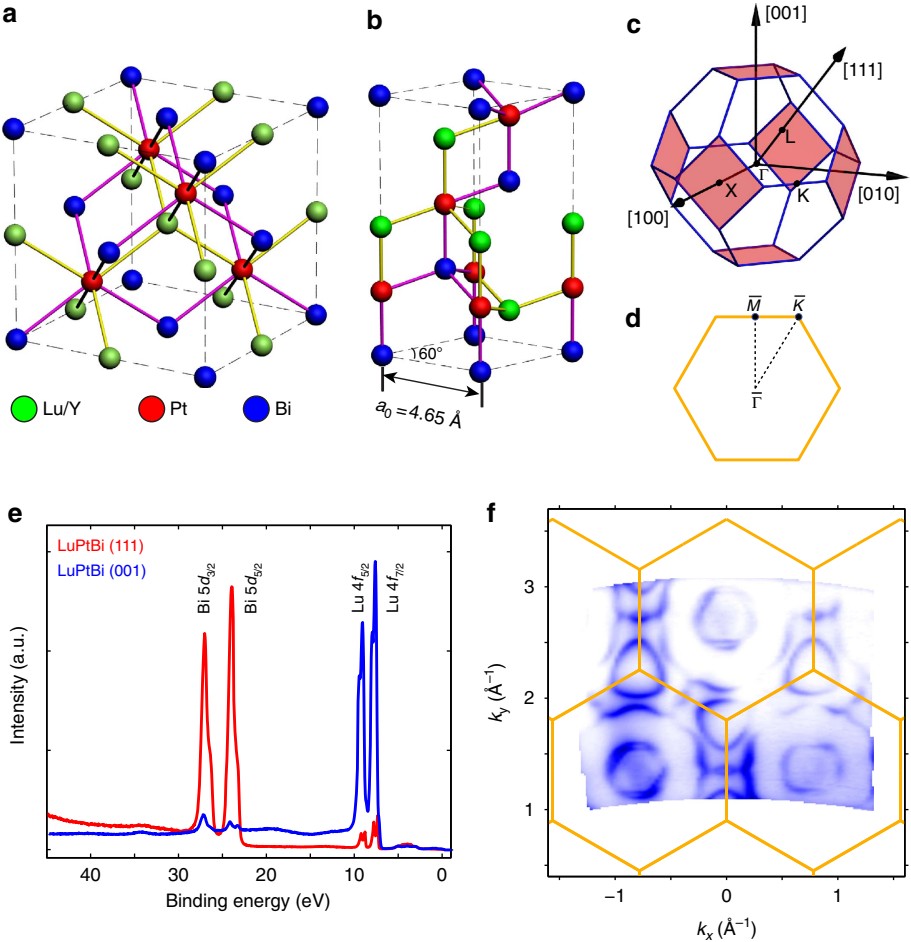

**Figure 1 | Crystal structure of LnPtBi and cleavage surface measured by ARPES. (a)** Crystal structure of half-Heusler alloy LnPtBi crystal shows a composite of zinc-blend and rocksalt lattices. **(b)** Unit cell of LnPtBi at the (111) cleavage surface shows the stacking of triangular Ln, Pt and Bi layers. $a_O$ is the in-plane lattice constant of the (111) surface unit cell. **(c)** Bulk BZ of LnPtBi with high symmetry points labelled. Arrows and shaded surfaces indicate the projection to [100], [010] and [001] directions. **(d)** Surface BZ in the [111] direction with the high symmetry points labelled. **(e)** Core level photoemission spectrum on LuPtBi (111) and (001) surfaces clearly shows the characteristic Lu 4f and Bi 5d doublets. These spectra are measured with 75 and 215 eV photons, respectively. **(f)** Broad FS map of LuPtBi covering five BZs, confirming the shape and size of the surface BZ (overlaid yellow hexagons) on the (111) cleave plane. The uneven intensity of the FS at different BZs results from the matrix element effect.

Such method could reveal TSSs clearly and avoid the trivial surface states by removing the dangling bond orbitals in the Hamiltonian (as can be clearly seen in Fig. 3b).

The combination of the two methods thus allows us to unambiguously identify the TSSs from other trivial dangling bond states (further experimental evidence of the non-trivial nature of the TSSs can be found in Supplementary Fig. 3 and Supplementary Note 2). As shown in Fig. 3a, all sharp dispersions (three Kramers pairs and one X shape state, labelled as SS1–SS3 and TSS respectively) are of surface origin and agree excellently with the observed band structures (Fig. 3c–f). Moreover, the three Kramers pairs (SS1, SS2 and SS3) are all absent in the result from method two (Fig. 3b) while the X shape band remains, illustrating their topologically trivial origin (that is, being trivial surface states due to dangling bonds), as opposite to the TSS shown in Fig. 3b.

The surface origin of both the TSS and SS1–SS3 in Fig. 3 can also be experimentally verified by performing the photon energy dependence photoemission measurement[30], as presented in Fig. 4 (also in Supplementary Fig. 4 and Supplementary Note 3 along the other high symmetry direction). In Fig. 4a, dispersions along $\bar{\Gamma}$–$\bar{K}$–$\bar{M}$ directions measured using a wide range of photon energies (50–75 eV) were plotted, the dispersions of TSS

and SS1–SS3 under all photon energies are identical (though the relative intensity can vary with photon energy due to the photoemission matrix element effect[30]). To further visualize the dispersion of these bands along $k_z$, we extract the momentum distribution curves (MDCs) at $E_F$ (cutting through SS1 and SS2) and 0.65 eV below $E_F$ (cutting through TSS, SS2 and SS3) and plot them as the function of photon energy (see Fig. 4b,c). Evidently, the peaks from TSS and SS1–SS3 bands show no $k_z$ dispersion as they all form straight vertical lines. Thus, the surface nature of these bands (TSS and SS1–SS3) are clearly established. By fitting the Dirac type X shape linear dispersion (Fig. 4d), we can extract the band velocity at the Dirac point as 2.37 eV Å ($3.59 \times 10^5$ m s$^{-1}$) and 3.13 eV Å ($4.74 \times 10^5$ m s$^{-1}$) along the $\bar{\Gamma}$–$\bar{K}$ and $\bar{\Gamma}$–$\bar{M}$ direction, respectively.

**Surface states on YPtBi (111) and LnPtBi (001) surface.** Similarly, for the other compound YPtBi, our calculation and measurements also agree well and both clearly show the TSS (Fig. 4e–g, also see Supplementary Fig. 5 and Supplementary Note 4 for the calculation). More measurements, as well as calculations on different cleaved surfaces (001) and different

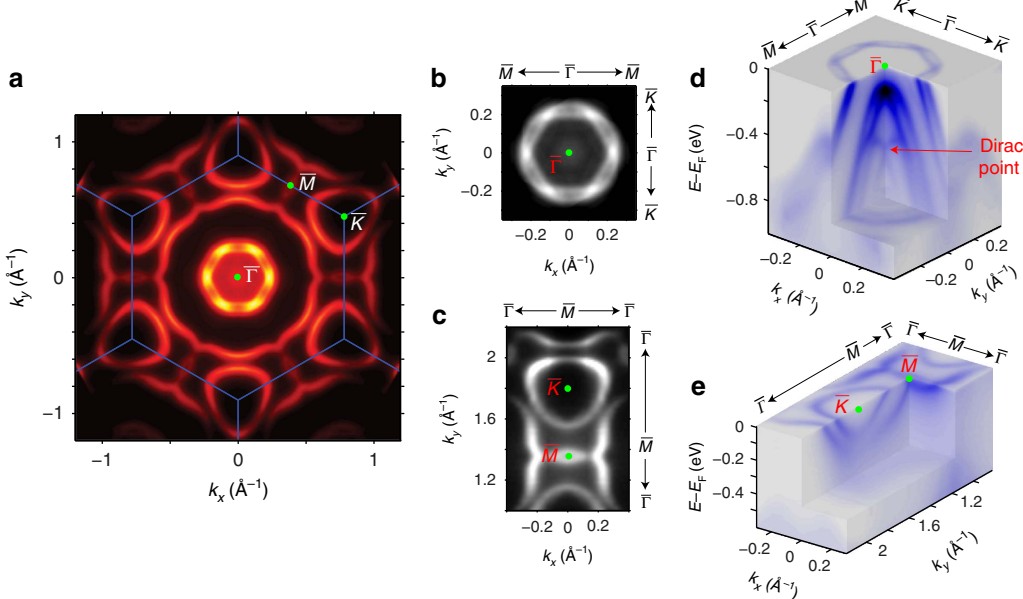

**Figure 2 | General electronic structure of LuPtBi (111) surface.** (**a**) FS maps of Bi-terminated LuPtBi (111) surface. Blue lines denote the surface BZ with high symmetry points labelled. The data has been symmetrized according to the crystal symmetry. (**b,c**) Zoom-in plot of FS map around the $\bar{\Gamma}$ point (**b**) and around the $\bar{M}$ and $\bar{K}$ points (**c**). (**d,e**) Plot of 3D electronic structure around the $\bar{\Gamma}$ point (**d**) and around the $\bar{M}$ and $\bar{K}$ points (**e**). All data were taken with 65 eV photons.

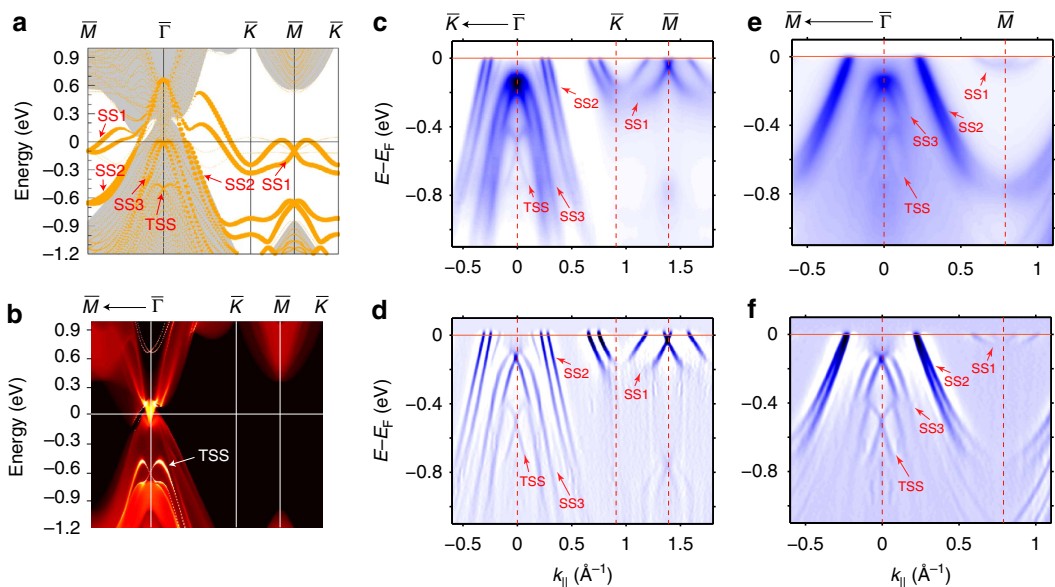

**Figure 3 | Observation of the metallic surface state and TSS on LuPtBi (111) surface.** (**a,b**) Calculated band structures of Bi-terminated LuPtBi (111) surface. (**a**) Result from a slab model calculation, in which the size of filled circles represent the projection to the Bi-terminated surface. Both topologically non-trivial surface state and metallic surface states are captured. (**b**) Results from a semi-infinite surface that is terminated by Bi. Only topologically non-trivial surface state is revealed by the calculation. (**c,d**) Photoemission intensity plot (**c**) and its second-derivative $\frac{\partial^2 I}{\partial E^2}$ plot (**d**) along the high symmetry $\bar{\Gamma}$–$\bar{K}$–$\bar{M}$ directions. (**e,f**) Photoemission intensity plot (**e**) and its second-derivative $\frac{\partial^2 I}{\partial E^2}$ plot (**f**) along the high symmetry $\bar{\Gamma}$–$\bar{M}$ directions. SS, topologically trivial metallic surface state due to the dangling bonds on sample surface. TSS, topologically non-trivial surface state. All data were taken with 65 eV photons.

termination layers (Bi or Ln terminations), are presented in Supplementary Figs 6 and 7 and Supplementary Note 5, all showing excellent agreements. Interestingly, in both compounds, the observed TSS coexist and overlap in energy with the bulk valence band (appear as broad dispersing background intensity in Figs 3 and 4), demonstrating its unusual robustness.

**Circular dichroism (CD)-ARPES of LuPtBi surface states.** CD-ARPES measurement on the Bi-terminated (111) surface of LuPtBi has also been carried out (see Fig. 5). As one can see from the cut along the $\bar{\Gamma}$–$\bar{M}$ direction, the photoemission spectra intensity under circularly polarized right (CR) or circularly polarized left (CL) photons only emphasizes one of the two

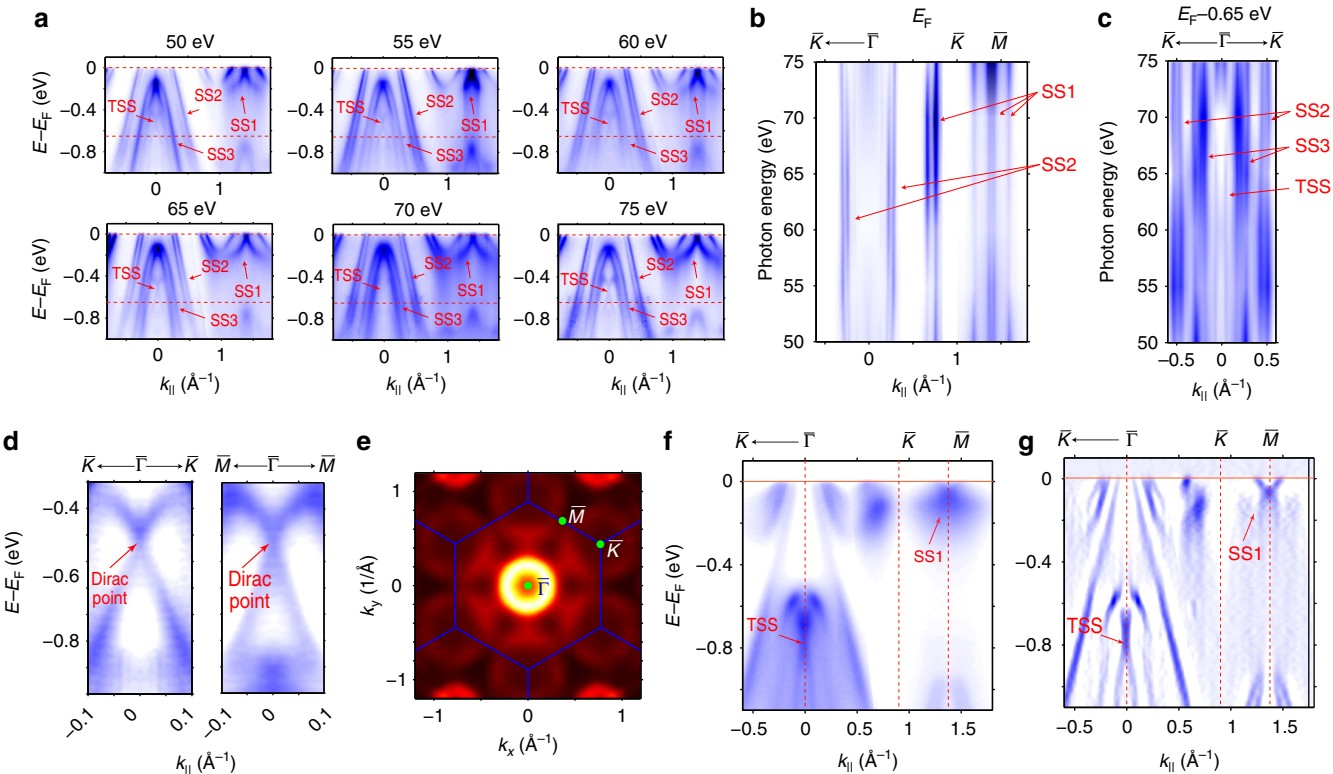

**Figure 4 | Photon energy dependence and Dirac-fermion behaviour of the TSS.** (**a**) Photoemission intensity plots along the high symmetry $\bar{\Gamma}$-$\bar{K}$-$\bar{M}$ direction with photon energies from 50 to 75 eV. Two red dotted lines marked the energy where individual MDC is taken to generate the plots in **b**,**c**. (**b**,**c**) Intensity plot of the MDCs taken at $E_F$ (**b**) and $E_F$—0.65 eV (**c**) at different photon energies. The MDC peak positions of all the observed bands are labelled (SS1–SS3 and TSS). (**d**) Zoom-in intensity plots of the TSS along the $\bar{\Gamma}$-$\bar{K}$ direction and the $\bar{\Gamma}$-$\bar{M}$ direction. Dirac points of the TSS are labelled. Data are taken with 60 eV photons. (**e**–**g**) Measured electronic structure of (111) surface of YPtBi. (**e**) The FS mapping of (111) surface of YPtBi with the hexagonal BZ (overlaid blue lines). The data has been symmetrized according to the crystal symmetry. (**f**,**g**) Photoemission intensity (**f**) and its second-derivative $\frac{\partial^2 I}{\partial E^2}$ plot (**g**) along the high symmetry $\bar{\Gamma}$-$\bar{K}$-$\bar{M}$ direction. Data in **e**–**g** are taken with 70 eV photons at $T = 20$ K. SS, topologically trivial metallic surface state due to the dangling bonds on sample surface. TSS, topologically non-trivial surface state.

branches of SS3 and TSS bands, showing clear circular dichroism as reflected in the CR–CL difference spectrum (Fig. 5a and the peak analysis of the MDCs at various energies in Fig. 5c). The obvious dichroic photoemission results are in consistence with the non-degenerate spin configurations in SS3 and TSS, as indicated by the spin-resolved *ab initio* calculations (see Supplementary Fig. 8 in Supplementary Note 6). We note that due to the complexity of the asymmetry of measurement geometry and orbitals effect in CD-ARPES[36], we encourage future spin-ARPES measurements for a more quantitative measurement on the spin polarization of TSS and SS3 bands.

## Discussion

Our discovery of the unusual TSSs on these materials establishing them as first examples with non-trivial topological electronic structure showing unusual robustness in Heusler materials with great tenability due to the vast number of compounds in the family. The non-trivial topology in the electronic structure can also help the understanding of various exotic properties recently discovered in half-Heusler compounds, including the large magnetoresistance[29], chiral anomaly and Weyl fermions[37,38] and their unconventional superconductivity[39,40]. Moreover, the interplay between topological electronic structure and the rich properties in Heusler materials further makes them ideal platforms for the realization of novel topological effects

(for example, exotic particles including image monopoles[41] and axions[42]) and new topological phases (for example, strain induced topological phase transition[14] and topological superconductors).

## Methods

**Angle-resolved photoemission spectroscopy.** ARPES measurements on single crystals LnPtBi (Ln = Lu, Y) were performed at beamline 10.0.1 of the Advanced Light Source (ALS) at Lawrence Berkeley National Laboratory, USA and beamline I05 of the Diamond Light Source (DLS), UK. The measurement pressure was kept below $3 \times 10^{-11}/8 \times 10^{-11}$ Torr in ALS/DLS, and data were recorded by Scienta R4000 analysers at 20 K sample temperature at both facilities. The total convolved energy and angle resolutions were 16 meV/30 meV and 0.2°/0.2° at ALS/DLS, respectively. ARPES circular dichroism measurement was performed at BL5-4 of the Stanford Synchrotron Radiation Lightsource, SLAC national laboratory with 20 eV photons.

**Ab initio calculations.** To simulate a surface, a 54-atomic-layer thick slab model was used with a vacuum more than 10 Å to diminish the coupling between the top and bottom surfaces. The *ab initio* calculations were performed within the framework of the density functional theory and generalized gradient approximation[43,44]. In the bulk calculations, the density functional theory Bloch wave functions were projected to Wannier functions[35], Ln-*d*, Pt-*sd* and Bi-*p* atomic like orbitals. In a half-infinite surface model, we projected the Green function of the bulk to the surface unit cell and obtained the surface local density of states based on the Wannier functions.

We could also calculate the surface bandstructure from tight binding model on the Bi-terminated (111) surface of LuPtBi. The metallic surface states (Fig. 4, SS1–SS3) observed on the Bi-terminated LuPtBi (111) surface all exhibit large spin

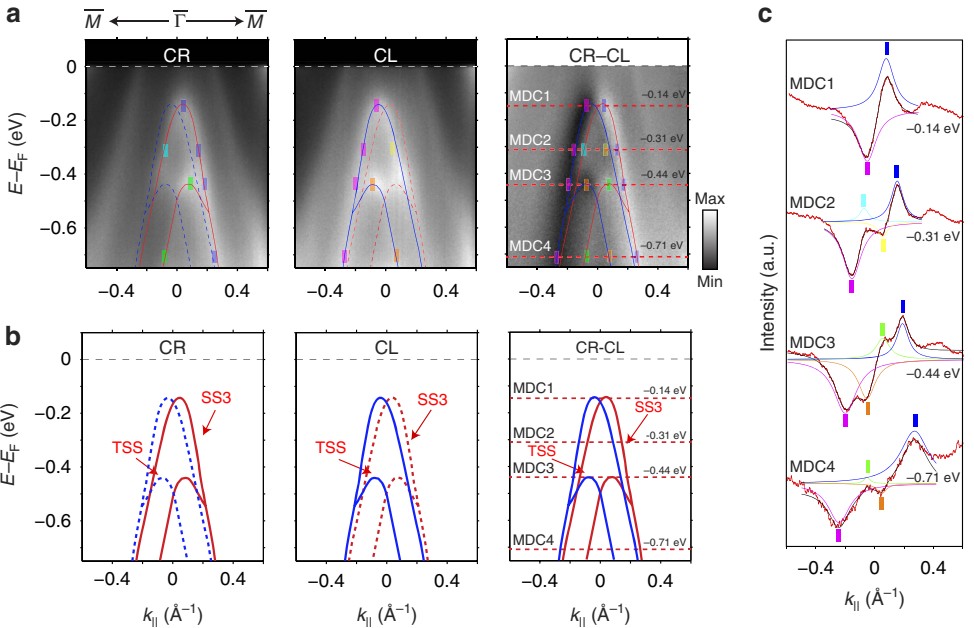

**Figure 5 | Circular dichroism of the metallic and TSS on LuPtBi (111) surface.** (**a**) Photoemission intensity around $\bar{\Gamma}$ along $\bar{\Gamma}$–$\bar{M}$ directions with the right and left circularly polarized (CR and CL) photons, showing clear asymmetry in the difference spectrum CR–CL. Dotted lines on the CR–CL spectrum represent the positions of the MDCs being analysed. Coloured marks label the position of the peaks fitted from the MDC curves in **c**. Red/Blue solid/dotted curves are the same as in **b**. (**b**) Red (Blue) solid curves give eye guides to the branch of bands which is enhanced with the CR (CL) photons, while blue (red) dotted curves give eye guides to the branch of bands which is suppressed with the CR (CL) photons. In the difference CR–CL spectrum both branches of SS3 and TSS could be seen. (**c**) MDCs analysis at the marked energy positions in **a**. Each fitted peak curve is drawn in colour, with the peak positions labelled with marks of same colour.

splitting, thus reflecting very large SOC strength. The surface state SS1 constitutes the FS, in particular, around the $\bar{K}$ and $\bar{M}$ points (Fig. 2). They are mainly composed by Bi-$p_z$ orbitals of the top Bi layer. The surface Bi atoms form a triangular lattice.

We wrote the effective tight-binding model for the SS1 surface states by considering the nearest neighbour hopping:

$$H = H^0 + H^R + H^{KM} \tag{1}$$

$$H^0 = -t\Sigma_{<ij> \sigma} c_{i\sigma}^+ c_{j\sigma} \tag{2}$$

$$H^R = \frac{\alpha}{2} \sum_{<ij> \sigma\sigma'} \left( \sigma \times \frac{\hat{e}_{ij}}{i} \right)_{z,\sigma\sigma'} c_{i\sigma}^+ c_{j\sigma'} \tag{3}$$

$$H^{KM} = \frac{\beta}{2i} \sum_{<ij> \sigma\sigma'} v_{ij} \sigma_z c_{i\sigma}^+ c_{j\sigma'} \tag{4}$$

(1) About the nearest neighbour hopping. We consider a triangle lattice as illustrated in Supplementary Fig. 9. We take the lattice parameter $a = 1$ and in-plane lattice vectors: $\mathbf{a_1} = (1, 0)$ and $\mathbf{a_2} = (1/2, \sqrt{3}/2)$. The six nearest sites of the site 0 are in unit of the lattice vectors: $\mathbf{R_1} = -\mathbf{R_4} = (1, 0)$, $\mathbf{R_2} = -\mathbf{R_5} = (1/2, \sqrt{3}/2)$ and $\mathbf{R_3} = -\mathbf{R_6} = (-1/2, \sqrt{3}/2)$.

For a single $p_z$ orbital on the triangular lattice

$$H^0(\mathbf{k}) = -t\left[\cos(k_x) + \cos\left(\frac{k_x}{2} + \frac{k_y\sqrt{3}}{2}\right) + \cos\left(-\frac{k_x}{2} + \frac{k_y\sqrt{3}}{2}\right)\right] \tag{5}$$

where $k_x$, $k_y$ are scaled by the unit of $\frac{2\pi}{a\sin(\frac{\pi}{3})} = \frac{4\pi}{\sqrt{3}a}$.

(2) The Rashba term can be written as

$$H^R = \sum_{<ij> \sigma\sigma'} H_{ij\sigma\sigma'} c_{i\sigma}^+ c_{j\sigma'} = \frac{\alpha}{2} \sum_{<ij> \sigma\sigma'} \left( \sigma \times \frac{\hat{e}_{ij}}{i} \right)_{z,\sigma\sigma'} c_{i\sigma}^+ c_{j\sigma'}, \tag{6}$$

where $H_{01}^R = -H_{04}^R = -\frac{\alpha}{2i}\sigma_y$, $H_{02}^R = -H_{05}^R = \frac{\alpha}{2i}\left(\frac{\sqrt{3}}{2}\sigma_x - \frac{1}{2}\sigma_y\right)$ and $H_{03}^R = -H_{06}^R = \frac{\alpha}{2i}\left(\frac{\sqrt{3}}{2}\sigma_x + \frac{1}{2}\sigma_y\right)$.

The Rashba Hamiltonian is

$$H^R(\mathbf{k}) = \alpha\left[-\sigma_y\sin(k_x) + \left(\frac{\sqrt{3}}{2}\sigma_x - \frac{1}{2}\sigma_y\right)\sin\left(\frac{k_x}{2} + \frac{k_y\sqrt{3}}{2}\right) + \left(\frac{\sqrt{3}}{2}\sigma_x + \frac{1}{2}\sigma_y\right)\sin\left(-\frac{k_x}{2} + \frac{k_y\sqrt{3}}{2}\right)\right] \tag{7}$$

(3) The Kane–Mele term $H^{KM}$. In the triangular lattice, the value of $v_{ij}$ depends on the hopping direction, as shown in Supplementary Fig. 9. We derived this term as

$$H^{KM}(\mathbf{k}) = \beta\sigma_z\left[\sin(k_x) - \sin\left(\frac{k_x}{2} + \frac{k_y\sqrt{3}}{2}\right) + \sin\left(-\frac{k_x}{2} + \frac{k_y\sqrt{3}}{2}\right)\right] \tag{8}$$

We fitted the surface band structure using the parameters $t = 0.35$, $\alpha = 0.5$ and $\beta = 0.1$ and well reproduced the *ab initio* surface states (Supplementary Figs 9 and 10). Without the Kane–Mele term, the band degeneracy cannot be lifted at the $\bar{K}$ point, although $\bar{K}$ is not time-reversal invariant.

Recently a similar SOC effect[45] has been employed to describe the surface states on the Tl/Si(111)–(1 × 1) surface[46]. Existence of strong Rashba effect and the unconventional SOC effect from the surface and the non-centrosymmetric bulk induces significant effect such as weak antilocalization and high mobility in transport measurements, and makes the LnPtBi a feasible candidate for spintronics applications.

**Data availability.** All relevant data are available on request, which should be addressed to Y.L.C.

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

## Acknowledgements

Y.L.C. acknowledges the support of the EPSRC Platform Grant (Grant No. EP/M020517/1) and Hefei Science Center CAS (2015HSC-UE013). C.F. acknowledges the financial support by the ERC Advanced Grant (No. 291472 'Idea Heusler'). J.J. acknowledges the support of the NRF, Korea through the SRC Center for Topological Matter (No. 2011-0030787). The ALS is supported by the Office of Basic Energy Sciences of the U.S. DOE under Contract No. DE-AC02-05CH11231. H.F.Y. acknowledges the finacial support from the Bureau of Frontier Sciences and Education, Chinese Academy of Sciences.

## Author contributions

Y.L.C. conceived the experiments. Z.K.L. and L.X.Y. carried out ARPES measurements with the assistance of J.J., H.F.Y., Y.Z. and S.-K.M. C.S. synthesized and characterized the bulk single crystals. S.-C.W. and B.Y. performed ab initio calculations. All authors contributed to the scientific planning and discussions.

## Additional information

**Competing financial interests:** The authors declare no competing financial interests.

