## [Peer Review File · Nature Communications]

Reviewers' comments:

Reviewer #1 (Remarks to the Author):

Ternary Heusler compounds have attracted great interests due to their rich physical properties and the possible topological nature predicted by calculations. The search for the topological nontrivial surface states in these materials is of great importance. This paper present clear measurements of the electronic structure of LnPtBi, providing convincing evidence for surface states below Fermi energy with "X" shape band dispersion, which was missing in previous ARPES reports. Combined with band structure calculations, the authors suggest that the band is topological nontrivial.

The paper is well written, and the data quality is high. I feel that their findings are sufficiently exciting and suggestive to warrant publication in Nature Communications, as long as the authors can adequately address the following questions/comments:

1. The main conclusion of this paper is that the "X" shape band is topologically nontrivial surface state. However, considering that the "X" shape band shows a back bending dispersion at the tip part, if the back bending dispersion is also surface state, the surface state can be eliminated by shifting the chemical potential, which sets doubts on the topological nontrivial nature of the band proposed in this paper. In the part D of the supplementary, the authors assume the back bending dispersion as the hybridization of surface state and bulk states, but did not provide any experimental evidence. At least, the authors should show the k_z dispersion of the band bending dispersion to support their assumption.
2. The current CD-ARPES results in the Supplementary Fig.S7 show poor data quality, making it hard to identify different bands, in contrast to the claimed "clear circular dichroism". To get convincing conclusion on the chiral texture for band "TSS" and "SS3", MDCs measured using "CR", "CL" and "CR+CL" photons should be compared. Moreover, as the spin resolved ARPES measurements have not been performed at the current stage, the CD-ARPES measurements which can hint on the spin configuration is very important, and should be presented in the main text.
3. The authors of Ref.27 observed a band contributed by surface state near Gamma with weak intensity, and they suggest that further ARPES experiments are required to explore whether this band is topological non-trivial or not. In the manuscript, the weak band is also clear in Fig.3c and Figs.4(a)(b)(f)(g); however, this band is not even mentioned. Could this band be explained by a Kramers pair of bands? The author should compare their results with the previous literatures and discuss on this open issue.

Reviewer #2 (Remarks to the Author):

The authors present study of electronic structure of rare earths half Heusler compounds. In addition to previously observed set of surface states, they found evidence for Dirac-like feature at the center of the Brillouin zone at -0.45 eV. Topological quantum materials are very active and exciting and still relatively new area of condensed matter physics. Rare earths half Heusler materials are particularly promising as they combine topological structure, superconductivity and heavy fermion behavior. The topic of this manuscript is therefore timely and interesting.

The data is of high quality and presence of Dirac-like feature at the center of the zone is established beyond doubt. Extensive photon energy dependent studies validate surface character of this feature. The manuscript is well written with clear motivation and nice data demonstrating clearly the

presence of the Dirac like feature. I support publication of this manuscript in Nature Communications. I would encourage authors to comment on other aspects of possible topological properties their data. One of the hallmarks of topologically non-trivial states, especially for any topological signatures in transport is presence of odd number of FS crossings. I understand that the feature reported here lies completely below E_f , but authors should examine this aspect of their data and comment in text.

Reviewer #3 (Remarks to the Author):

In this manuscript, Z. K. Liu and et.al. reported observation of unusual topological surface states in half-Heusler compounds LnPtBi ($\text{Ln}=\text{Lu}$ or Y), an interesting compound which is non-centrosymmetric and superconducting at low temperature. These properties, if combined with non-trivial topological states, are indeed attractive and have been drawing recent attentions (for example, Weyl semimetal without inversion symmetry or topological superconductors). Half-Heusler compounds have been theoretically predicted to be topologically non-trivial, and some of the compounds ($\text{Ln}=\text{Lu}$, Dy , Gd) have been measured before (Ref.27), but with no clear conclusions. In this manuscript, however, by careful ARPES measurement and detailed comparison with ab-initio calculations, Liu et.al observed clear surface states as well as a non-trivial Dirac cone well below E_f at gamma point with high resolutions. This is very nice and provide experimental support to the theory and prediction. I should say this is a piece of nice work with beautiful data and careful analysis that are convincing, however, I do not think it is novel enough to justify its publication on Nature Commun. My judgement is based on the following considerations.

(1) The compounds are actually metallic, namely, they are NOT topological insulators. In principle, in the absence of gap, the \mathbb{Z}_2 topological property is ill-defined (except topological metal, where surface Fermi arcs are expected). In this case, the so-called topological surface states, if any, are not protected.

(2) The surface states observed at Fermi levels are actually all trivial. The only claimed Dirac cone (TSS) is located well below Fermi level. This is again unattractive, because topological surface states are interesting only if they are located within a band-gap and crossing the Fermi level. Dirac cone type surface states away from Fermi level may widely exist in most of materials (if not all).

(3) The above two weak points make the compounds "boring", although their data are beautiful. This fact also weakens the possible connection, as claimed by authors, between the topology and the specially interesting non-centrosymmetry as well as superconductivity in this family of compounds. In general, we expect the superconductivity is related to the Fermi level and Fermi surfaces (which are all trivial in this case as observed).

In summary, I believe the manuscript provides interesting and convincing information, particularly for half-Heusler compounds, but its importance may not satisfy the criteria of Nature Commun. I therefore suggest author to expand the paper and submit it to specialized magazines.

REVIEWERS' COMMENTS:

Reviewer #1 (Remarks to the Author):

The authors have made substantial revisions in respond to the comments and questions raised by the reviewers. I recommend accepting the revised manuscript to be published in Nature Communications.

Reviewer #2 (Remarks to the Author):

I find the response the authors to both reports satisfactory and revised version of the manuscript suitable for publications.

Reviewer #3 (Remarks to the Author):

Authors have made comprehensive response to my previous report. I think the response is good and reasonable. As I have explained in my previous report, this paper is nice and provides convincing data to support the non-trivial topological nature of LnPrBi class of materials. On the other hand, however, it is indeed difficult to justify its novelty. First, for topological insulators, the existence of bulk gap is important for real transport measurement. Otherwise, the topological surface states, if any, play minor roles for the global properties. Second, for topological semimetals, the bulk states should play more important roles than the surface fermi arcs. In my opinion, nowadays, there are too many ARPES studies, which confirm theoretical calculations and report the existence of non-trivial surface states. These studies are good, contribute the field, but not as significant as some years ago (when the field just started). Nowadays, the most important issue (from the viewpoint of materials) is to find better materials which can be well manipulated and controlled. I therefore leave the final judgement up to the editor.

Response to Reviewer's comments:

Reviewer #1:

Ternary Heusler compounds have attracted great interests due to their rich physical properties and the possible topological nature predicted by calculations. The search for the topological nontrivial surface states in these materials is of great importance. This paper present clear measurements of the electronic structure of LnPtBi, providing convincing evidence for surface states below Fermi energy with "X" shape band dispersion, which was missing in previous ARPES reports. Combined with band structure calculations, the authors suggest that the band is topological nontrivial.

The paper is well written, and the data quality is high. I feel that their findings are sufficiently exciting and suggestive to warrant publication in Nature Communications, as long as the authors can adequately address the following questions/comments:

Authors' response:

We thank the reviewer for acknowledging the importance of this work and the high quality of our data. We also thank the reviewer for the helpful suggestion. As suggested, we present further analysis below (and in the revised manuscript) to address the questions/comments from the reviewer.

1. The main conclusion of this paper is that the "X" shape band is topologically nontrivial surface state. However, considering that the "X" shape band shows a back bending dispersion at the tip part, if the back bending dispersion is also surface state, the surface state can be eliminated by shifting the chemical potential, which sets doubts on the topological nontrivial nature of the band proposed in this paper. In the part D of the supplementary, the authors assume the back bending dispersion as the hybridization of surface state and bulk states, but did not provide any experimental evidence. At least, the authors should show the k_z dispersion of the band bending dispersion to support their assumption.

Authors' response:

Thanks for the reviewer's suggestion to add more discussions and k_z dispersion of the back-bending dispersion. We have added the k_z dispersion of the "X" shape band at different binding energy, and confirm that the back-bending part indeed shows considerable k_z -dispersion (as opposed to the linearly dispersing part which does not), thus providing clear experimental evidence that the back-bending part indeed hybridize and merge into the bulk band. The related discussions have been added as the new Part B in the supplementary information of the revised manuscript.

In Fig. R1 below (we also add this figure into the Supplementary Information as new Fig. S2), we chose two momentum distribution curves (MDCs) that cut through the "X" shape dispersions at different binding energy to demonstrate their different k_z dispersions (MDC1 cuts through the back-bending part near the band top ($E_b=0.41\text{eV}$) and MDC2 cuts through the linear dispersion part ($E_b=0.67\text{eV}$), respectively).

For better illustration, we mark the MDC peaks related to back-bending part (blue circles, see Fig. R1c(i)), indeed they show clear variation with different photon energy. Such behavior shows clear difference from the k_z dispersion of MDC2 (which comes from the linearly dispersing part) where the green circles (indicating the MDC peaks, see Fig. R1c(ii)) show no photon energy dependent variation.

The k_z dispersion shown in Fig. R1 (and in current Supplementary Information Fig. S2) thus provides clear experimental evidence of the hybridization between the back-bending part of the TSS dispersion and the bulk state.

Fig. R1 Photon energy dependence of different part of TSS. (a)(b) Dispersions along high symmetry direction ($\bar{\Gamma}$ - \bar{K}) with 65eV (a) and 55eV (b) photon energy, respectively, panels (ii) in (a)(b) show the zoom-in plots of the “X” shape TSS dispersion with details. Orange and magenta dashed lines in panels (ii) of (a)(b) indicate the two MDCs at different parts of the “X” shape dispersion which we use for photon energy dependent measurement in (c). (c), Intensity plot of the photon energy dependent measurements of MDCs at 0.41eV(i) and 0.67eV(ii) binding energy. Blue and green circles represent the fitted peak positions on each of the MDC. The orange and magenta dashed lines show the MDCs from panels (ii) of (a) and (b), respectively. TSS: topological surface

2. The current CD-ARPES results in the Supplementary Fig.S7 show poor data quality, making it hard to identify different bands, in contrast to the claimed "clear circular dichroism". To get convincing conclusion on the chiral texture for band "TSS" and "SS3", MDCs measured using "CR","CL" and "CR+CL" photons should be compared. Moreover, as the spin resolved ARPES measurements have not been performed at the current stage, the CD-ARPES measurements which can hint on the spin configuration is very important, and should be presented in the main text.

Authors' response:

We thank the reviewer for the suggestions on making better comparisons on the CD-ARPES data (as these measurements were carried out at low energy (20eV), the photoemission cross section of LuPtBi under such energy photons caused relatively worse statistics compared to other measurements in the manuscript).

In Fig. R2 (also the new Fig.5 added in the main text of the revised manuscript) we show more detailed CD analysis including the fitting to four MDCs cutting through the TSS and SS3 bands under circular right (CR) and circular left (CL) photons, as well as their difference spectrum (CR-CL). The circular dichroism of both SS3 and TSS is more apparent in the CR-CL spectrum (see Fig. R2(a)(iii)).

The fitting of the four MDCs in Fig. R2(c) further proves the chiral texture for TSS and SS3 bands. In MDC1 that cuts through the band top of SS3 (0.14eV below E_F), we observed a negative peak (labeled by purple mark) and a positive peak (labeled by the blue mark), corresponding to the two branches of the SS3 which has stronger intensity with “CR” and “CL” photons, respectively. Going towards higher binding energy, the single peak of each branch evolves into two, as seen in the MDC2 at 0.31eV below E_F : purple peak splits into purple and yellow while blue peak splits into blue and cyan (the fitted peaks from the same branch show the same sign of amplitude).

As for the TSS, the fittings to the MDC3 at $E-E_F=-0.44$ eV and MDC4 at $E-E_F=-0.71$ shows one branch of the ‘X’ shape is stronger with “CR” photons (green marks) thus appears as a positive peak in the “CR-CL” spectrum; while the other branch is stronger with “CL” photons (orange peaks) and appears as a negative peak.

Therefore, our ARPES measurement indeed shows clear circular dichroism signal from both SS3 and TSS band: each branch of the TSS and SS3 is suppressed with one circularly polarized photons and enhanced with the other.

We have added Fig. R2 as the new Fig. 5 into the main text of the revised manuscript together with the corresponding discussions.

Fig. R2 Circular dichroism of the metallic and topological surface state on Bi-terminated LuPtBi (111) surface. (a) Photoemission intensity around $\bar{\Gamma}$ along $\bar{\Gamma}-\bar{M}$ directions with the right (i) and left (ii) circularly polarized (CR and CL) photons, showing clear asymmetry in the difference spectrum CR-CL (iii). Dotted lines on (iii) represent the positions of the MDCs being analyzed. Colored marks label the position of the peaks fitted from the MDC curves in (c). Red/Blue solid/dotted curves are the same as in (b). (b) Red (Blue) solid curves give eye guides to the branch of bands which is enhanced with the CR (CL) photons, while blue (red) dotted curves give eye guides to the branch of bands which is suppressed with the CR (CL) photons. In the difference spectrum both branches of SS3 and TSS could be seen. (c) MDCs analysis at the marked energy positions in (a)(iii). Each fitted peak curve is drawn in color, with the peak positions labeled with marks of same color.

3. The authors of Ref.27 observed a band contributed by surface state near Gamma with weak intensity, and they suggest that further ARPES experiments are required to

explore whether this band is topological non-trivial or not. In the manuscript, the weak band is also clear in Fig.3c and Figs.4(a)(b)(f)(g); however, this band is not even mentioned. Could this band be explained by a Kramers pair of bands? The author should compare their results with the previous literatures and discuss on this open issue.

Authors' response:

We thank the referee for the careful reading and pointing that out. Indeed there is a weak band near $\bar{\Gamma}$ next to the SS2 band (see Fig. R3(a) below, indicated by the green arrow). The position and shape of this band is similar to the band reported in Ref. 27 (currently Ref. 29 in the revised manuscript), which was suggested in Ref. 27 as a Rashba type surface state. However, the fact that this band only shows sharp single branch dispersion make it very different from other Kramers pair in this materials, such as SS2 and SS3 nearby that both show clear (and large) spin-splitting.

To investigate the origin of this band, we first carry out photon energies dependent measurement and study its k_z -dispersion (Fig. R3(b)(c)). As clearly shown in Fig. R3(c), the position of the Fermi surface crossing and shape does not show much variation along k_z (Fig. R3(c)), showing its surface origin. However, this band also shows considerable difference from other Rashba type surface states nearby, such as adjacent SS2 and SS3 bands in that: (1) this band does not come in pairs and (2) the dispersion has much weaker intensity comparing to SS3 and SS2 in ALL measurements. These differences indicate that this band is not typical Rashba split

Fig. R3 Analysis of the E_F crossing bands around $\bar{\Gamma}$. (a) Zoomed-in plot of the $\bar{\Gamma}$ - \bar{K} cut close to E_F . Different bands are labeled separately. The Fermi crossings are identified by fitted peaks on the MDC (the above panel). (b) Plot of the $\bar{\Gamma}$ - \bar{K} cut measured at various photon energies. Different bands are labeled. (c) Intensity plot of the stack MDCs at E_F extracted from the cuts in (b). For each MDC, the positions of each peaks fitted are labeled by the colored marks. SS: topologically trivial surface state. RS: surface resonant state.

surface state (such as SS2 and SS3).

In order to understand the nature of this band, we carry out *ab-initio* calculations under different conditions.

From Fig. R4(a)-(c), the calculations show the evolution of the band dispersion projected to thin to thick (2-4 layers) surface slabs. While the SS2 and SS3 bands show similar intensity from Fig. R4(a)(i)-(c)(i), the weak band (highlighted by the transparent red ovals in the Fig. R4(a)(ii)-(c)(ii)) under investigation shows clear difference: in Fig. R4(a)(ii), this band is very weak, but with the increase of the slab thicknesses, the spectra weight of this band clearly increases (Fig. R4(b)(ii), Fig.

R4(c)(ii)). Such enhancement of spectral weight with thicker slabs, together with its surface nature we demonstrated in Fig. R3, suggest that it is a surface resonant state (a mixture of surface and bulk states), which satisfactorily explains the absence of the k_z dependence and its different features from Rashba type surface states (such as SS2 and SS3).

The discussions above have also been added into the Supplementary Information of the revised manuscript.

Fig. R4 Calculated bandstructure of Bi-terminated LuPtBi (111) surface along high symmetry cuts. (a-c) (i) Results from a slab model calculation when projected to 2-, 3- and 4- layer. The size of filled circles represent the contribution from the surface. Different bands are labeled. (ii) Zoom-in plot of the calculation results from (a-c)(i). Different bands are labeled. Red ovals indicate the band between SS2 and SS3. (d) Plot of the measured spectrum of the same range. Different bands are labeled. SS: topologically trivial surface state. TSS: topologically non-trivial surface state. RS: surface resonant state.

Response to Reviewer's comments:

Reviewer #2:

The authors present study of electronic structure of rare earths half Heusler compounds. In addition to previously observed set of surface states, they found evidence for Dirac-like feature at the center of the Brillouin zone at -0.45 eV. Topological quantum materials are very active and exciting and still relatively new area of condensed matter physics. Rare earths half Heusler materials are particularly promising as they combine topological structure, superconductivity and heavy fermion behavior. The topic of this manuscript is therefore timely and interesting.

The data is of high quality and presence of Dirac-like feature at the center of the zone is established beyond doubt. Extensive photon energy dependent studies validate surface character of this feature. The manuscript is well written with clear motivation and nice data demonstrating clearly the presence of the Dirac like feature. I support publication of this manuscript in Nature Communications. I would encourage authors to comment on other aspects of possible topological properties their data. One of the hallmarks of topologically non-trivial states, especially for any topological signatures in transport is presence of odd number of FS crossings. I understand that the feature reported here lies completely below E_F , but authors should examine this aspect of their data and comment in text.

Authors' response:

We thank the reviewer for acknowledging the importance of this work and the high quality of our data. We also thank the reviewer for the helpful suggestions.

Regarding the non-trivial topological surface state (TSS) we report in the manuscript, although it is located ~ 500 meV below E_F and does not contribute directly to the transport properties, its existence directly proves the non-trivial topological invariant in the half-Heusler alloy – this could help to explain the many interesting/exotic properties discovered in this compound (and possibly also other half-Heusler compounds), such as the high mobility electrons and large magnetoresistance (Z. P. Hou et al., Phys. Rev. B 92, 235134(2015)), chiral anomaly and Weyl fermions under magnetic field (M. Hirschberger et al., arXiv:1602.07219 and C. Shekhar et al. arXiv:1604.01641) and unconventional superconductivity which hosts nodal gap structure (H. Kim et al., arXiv:1603.03375v2) and possible spin-triplet pairing mechanism (Y. Nakajima et al., Science Advances 1, e1500242 (2015)).

Moreover, the TSS could become more relevant after proper manipulation of the doping level and the band structure in this compound: By tuning of the Fermi level, one would be able to move the E_F close to TSS and have the topological surface electrons contribute to the superconductivity pairing, and thus achieve one possible scenario of topological superconductivity. In addition, there have been theoretical works demonstrating that half-Heusler compounds will become a topological insulator with the TSS sitting in the bandgap under uniaxial strain (D. Xiao et al., Phys. Rev. Lett. 105, 096404(2010)); or become Weyl semimetals under compressive strain (J. W. Ruan et al., Nat. Commun. 7:11136 (2016)). Therefore, our identification of the non-trivial topology in half-Heusler compounds is an important milestone which will encourage the exploration for these new topological phases. Besides, as the TSS and its evolution would be important in these new topological phases (TIs or Weyl semimetals), our discovery of the TSS thus provides an important indicator for the identification, bandstructure engineering and the observation of topological phase transitions.

As suggested by the reviewer, we have also added the discussions into the manuscript (page 7, paragraph 2).

Response to Reviewer's comments:

Reviewer #3:

In this manuscript, Z. K. Liu and et.al. reported observation of unusual topological surface states in half-heusler compounds LnPtBi ($\text{Ln}=\text{Lu}$ or Y), an interesting compound which is non-centrosymmetric and superconducting at low temperature. These properties, if combined with non-trivial topological states, are indeed attractive and have been drawing recent attentions (for example, Weyl semimetal without inversion symmetry or topological superconductors). Half-heusler compounds have been theoretically predicted to be topologically non-trivial, and some of the compounds ($\text{Ln}=\text{Lu}$, Dy , Gd) have been measured before (Ref.27), but with no clear conclusions. In this manuscript, however, by careful ARPES measurement and detailed comparison with ab-initio calculations, Liu et.al observed clear surface states as well as a non-trivial Dirac cone well below E_f at gamma point with high resolutions. This is very nice and provide experimental support to the theory and prediction. I should say this is a piece of nice work with beautiful data and careful analysis that are convincing, however, I do not think it is novel enough to justify its publication on *Nature Commun*. My judgement is based on the following considerations.

Authors' response:

We first thank the reviewer for acknowledging the quality and value of our work. However, with due respect, we do not agree with the reviewer's judgment that the work is not "novel enough". Please allow us to elaborate below the significance of Heusler topological materials below, why they are different from topological insulators, and their influence/implication to many important and exciting scientific problems – all of these make the discovery of the non-trivial topology in these compounds really important and exciting, thus worth the dissemination to broad audience of *Nature Communications*:

(1) Half-Heusler alloys represent a big family compounds that interest physicists, chemists and material scientists with many intriguing properties (including unusual magnetism and superconductivity) which need to be understood. In particular, they have drawn considerable research attentions recently due to the discovery of various exotic properties, such as:

a) The high mobility and large magnetoresistance (Z. P. Hou et al., *Phys. Rev. B* 92, 235134(2015)),

b) Chiral anomaly and Weyl fermions under magnetic field (M. Hirschberger et al., arXiv:1602.07219 and C. Shekhar et al. arXiv:1604.01641)

c) Unconventional superconductivity which hosts nodal gap structure (H. Kim et al., arXiv:1603.03375v2) and

d) Possible spin-triplet pairing mechanism (Y. Nakajima et al., *Science Advances* 1, e1500242 (2015)).

Many of these new properties strongly indicate or require that the topological electronic structure to plays an important role in these compounds, yet a direct evidence of their non-trivial topological nature is still lacking. Moreover, the fact that Ref 27 (currently Ref 29 in the revised manuscript) gave a negative conclusion (due to the data quality) makes the situation more confusing, thus a clarification of is urged. Our work, for the first time, clearly demonstrated the topological surface states in

$LnPtBi$, thus unambiguously proves the non-trivial topological nature of these half-Heusler compounds.

(2) In addition to the observation of the topological surface states, the significance of our work goes beyond. As these topological surface states are unusual: Naively one would assume that the topological surface states must appear inside the bulk gap (this was part of the reason why they were overlooked in previous experimental and theoretical study), nevertheless our results clearly show that they can exist inside the inverted band gap – which is not necessarily the normal gap between conduction and valence bands – and can (surprisingly!) survive the hybridization with the bulk states, thus the topological surface states are not as vulnerable as one previously thought and must live in the normal band energy gap.

(3) Besides the topological surface states, our work confirms the inverted bulk band structure, - which is the prerequisite to understand the unusual superconductivity in these compounds, for instance, to interpret the unconventional pairing mechanism.

(4) The revelation of the electronic structure and topological nature of the half-Heusler compounds can further provide important guidance for tailoring the band structures and engineering novel topological material/phases. For example, by properly tuning the Fermi level, we could move E_F to intersect the topological surface states and have the topological surface electrons contribute to the superconductivity pairing, thus achieve one possible scenario of topological superconductivity. In addition, there have been theoretical works demonstrating that half-Heusler compounds will become a topological insulator with the TSS sitting in the bandgap under uniaxial strain (D. Xiao et al., Phys. Rev. Lett. 105, 096404(2010)); or become Weyl semimetals under compressive strain (J. W. Ruan et al., Nat. Commun. 7:11136 (2016)).

Therefore, our identification of the non-trivial topology in half-Heusler compounds, for the first time, is an important milestone which will provide a solid foundation and further encourage the exploration for novel topological phases in this exciting big family of materials.

Below we would like to respond to the reviewer's technical comments in details:

(1) The compounds are actually metallic, namely, they are NOT topological insulators. In principle, in the absence of gap, the Z2 topological property is ill-defined (except topological metal, where surface Fermi arcs are expected). In this case, the so-called topological surface states, if any, are not protected.

Authors' response:

Although Heusler compounds (and their sister compound HgTe) are indeed gapless in the bulk band structure, they have been demonstrated to be topologically equivalent to topological insulators, as is commonly accepted. As an example, there have been theoretical work (e.g. PRL 105, 096404 (2010)] that showed the Z2 index can be well defined as long as an infinitesimal gap opens due to weakly symmetry breaking. Therefore, topological surface states are indeed protected by the Z2-type inverted bulk band structure.

(2) The surface states observed at Fermi levels are actually all trivial. The only claimed Dirac cone (TSS) is located well below Fermi level. This is again unattractive, because topological surface states are interesting only if they are located within a

band-gap and crossing the Fermi level. Dirac cone type surface states away from Fermi level may widely exist in most of materials (if not all).

Authors' response:

We do not agree with the reviewer on this point, and guess that the reviewer's comment is about the topological insulators – if the reviewer considers that the topological surface states are only interesting when they are in the bulk gap – then he/she is dismissing all the importance and exciting recent researches on topological Dirac and Weyl semimetals discovered recently which do not have a bulk gap.

Indeed, this work is about the topological semimetals, and that the TSSs are below the Fermi energy does not necessarily mean they are unattractive or uninteresting. As we have stated before, the existence of TSSs is originated and protected by the non-trivial topology, thus holding all the important properties (e.g., spin-momentum locking), similar to the TSSs that locate inside the band-gap of topological insulators. Moreover, in addition to the TSSs, the non-TSS surface states also exhibit rather interesting Rashba-type spin splitting, which is appealing for spintronics applications. In fact, both TSSs and non-TSS surface states in these half-Heusler compounds demonstrate attractive physics and exotic properties, making topological Heusler compounds a rich platform to test fundamental theory and explore future applications.

Finally, we would like to stress that the main message of the manuscript is not only limited to the discussion on the fascinating properties of either TSS or other non-topological surface states. Rather, by showing the existence of TSS, we established the topological non-trivial nature of the half-Heusler compounds we measured, which is essential for the understanding of many novel properties recently discovered (e.g., superconductivity with unconventional pairing and nodal gap structure, chiral anomaly and large magnetoresistance). Also, the complete band structures obtained in this work can also serve as a guide for tailoring the bandstructure for better properties (e.g. our measurement suggest that if the sample is p-doped, one could shift E_F to intersect the TSS, etc.).

(3) The above two weak points make the compounds "boring", although their data are beautiful. This fact also weaken the possible connection, as claimed by authors, between the topology and the specially interesting non-centrosymmetry as well as superconductivity in this family of compounds. In general, we expect the superconductivity is related to the Fermi level and Fermi surfaces (which are all trivial in this case as observed).

Authors' response:

As have been discussed above, that the absence of the bulk gap and the position of the TSS does NOT make the compounds “boring” (which have also been demonstrated by the intensive studies on topological semimetals recently).

In addition, it is not true that the connection between the topology and non-centrosymmetry is weakened by the absence of the gap, since the definition of the Z_2 invariant is irrelevant to the gap size as long as the band inversion occurs.

As for the superconductivity, although the TSS electrons do not directly participate the superconducting pairing (as they are 0.5eV below E_F), they demonstrate the nontrivial topology in these compounds, and the inverted bulk bands' electrons can contribute to superconducting pairing. In fact, a very recent experimental report suggests that the pairing takes place between the $j = 3/2$ fermions of the inverted p-like Γ_8 band and thus could host unconventional p-wave symmetry (H. Kim et al.,

arXiv:1603.03375v2), which shows the importance of our work that give the first demonstration that these compounds are topologically non-trivial.

In summary, I believe the manuscript provides interesting and convincing information, particularly for half-heusler compounds, but its importance may not satisfy the criteria of Nature Commun. I therefore suggest author to expand the paper and submit it to specialized magazines.

Authors' response:

The reviewer may consider that only finding the TSSs inside the bulk gap (such as in a topological insulator) is interesting. But as discussed above, we cannot agree with this opinion, in fact, the intensive studies on the topological Dirac and Weyl semimetals recently (e.g. Science 343, 864 (2014), Science 349, 613 (2015), Phys. Rev. X 5, 011029 (2015), Nature 527,495 (2015), Science 351, 1184 (2016), etc.) have shown the importance of the topological phase in non-bulk insulating compounds.

The discovery of the non-trivial topological nature in half-Heusler compounds, as discussed above, will be an important milestone that provide a solid foundation for the exploration of novel topological phases and the study of the rich and intriguing phenomena in this big family of materials, and thus worth the dissemination to broad audience of *Nature Communications*.

Response to Reviewer's comments:

Reviewer #1 (Remarks to the Author):

The authors have made substantial revisions in respond to the comments and questions raised by the reviewers. I recommend accepting the revised manuscript to be published in Nature Communications.

Authors' response:

We thank the reviewer for acknowledging our effort in improving the manuscript and are glad that the questions raised by the reviewer have been addressed!

Reviewer #2 (Remarks to the Author):

I find the response the authors to both reports satisfactory and revised version of the manuscript suitable for publications.

Authors' response:

We are glad that our revised manuscript is approved by the reviewer and consider our response responded the reviewers' questions satisfactorily!

Reviewer #3:

(i) Authors have made comprehensive response to my previous report. I think the response is good and reasonable. As I have explained in my previous report, this paper is nice and provides convincing data to support the non-trivial topological nature of LnPtBi class of materials. On the other hand, however, it is indeed difficult to justify its novelty.

Authors' response:

We appreciate the reviewer's acknowledgement that "this paper is nice and provides convincing data to support the non-trivial topological nature of LnPtBi class of materials"; and his/her satisfaction on with our previous reply: "Authors have made comprehensive response to my previous report. I think the response is good and reasonable."

As to the novelty of discovering the topological states in Heusler compounds, we would like to explain a bit more here. Half-Heusler alloys have long been a fascinating material family for condensed mater physicists due to the various intriguing properties (such as unusual magnetism and superconductivity). More recently, it further drew considerable research attentions due to the discovery of various exotic properties, including high mobility and large magnetoresistance, chiral anomaly and Weyl fermions under magnetic field, unconventional superconductivity which hosts nodal gap structure and possible spin-triplet pairing mechanism. As these properties strongly hint the non-trivial topological property in the electronic structure of these compounds, the lack of evidence of the non-trivial topological electronic structure in these materials is intriguing.

This work, for the first time, clearly demonstrated the topological surface states in LnPtBi, thus unambiguously proves the non-trivial topological nature of these half-Heusler compounds and thus open up the study of the topological electronic structures in this big family of materials which could be a rich ground for searching novel physical phenomena and exotic new quantum states. Based on these considerations, we think our study is novel and important for the development of the field.

(ii) First, for topological insulators, the existence of bulk gap is important for real transport measurement. Otherwise, the topological surface states, if any, play minor

roles for the global properties. Second, for topological semimetals, the bulk states should play more important roles than the surface fermi arcs.

Authors' response:

In terms of regular electric transport properties of topological semimetals (compared to topological insulators with bulk gap), indeed the surface state plays a relatively smaller role compared to the bulk states. However, the existence of the topological surface state demonstrates the non-trivial topology of the BULK electronic structure, which itself can play important roles in the transport properties (e.g. the chiral anomaly in topological semimetals are the result of the non-trivial bulk, rather than surface electrons). The rich unusual properties in half-Heusler compounds, combined with the topologically nontrivial electronic structure thus provide excellent materials base for the exploration of other more exotic quantum phases, such as topological superconductivity. Thus the implications of the non-trivial topological electronic structure in Heusler compound has much more profound aftermath, in addition to the contribution to the regular electric transport.

(iii) In my opinion, nowadays, there are too many ARPES studies, which confirm theoretical calculations and report the existence of non-trivial surface states. These studies are good, contribute the field, but not as significant as some years ago (when the field just started). Nowadays, the most important issue (from the viewpoint of materials) is to find better materials which can be well manipulated and controlled. I therefore leave the final judgement up to the editor.

Authors' response:

We agree with the reviewer that it is important to find better materials which can be well manipulated and controlled, as many of the physicists, chemists and material scientists (including us) are working, for examples, on the search of topological insulators with larger gap, better physical and chemical properties.

However, on the other hand, it is also important to search for novel topological state other than topological insulators (in the case of this work, topological semimetals), for the exploration of more unusual physical phenomena. The Heusler compounds, as we have explained in the manuscript, is a family of compounds that host many exotic properties and awaits to be understood and explored. This kind of exploration will advance our research forefront and expand our research into new territories.

We're glad that the powerful ARPES technique, which was used to the successful discovery of 3D topological insulators a few years ago, has been improved (e.g. with broader k_z range scanning for more accurate 3D band structure identification) and again successfully applied in the exploration of even more unusual topological quantum phases. We will keep improving the ARPES technique itself (e.g. with the addition of spin-resolution, super high spatial and time resolution) and make it an even more powerful experimental tool for the comprehensive study of electronic structures in novel quantum materials.